# Mesostats—A multiplexed, low-cost, do-it-yourself continuous culturing system for experimental evolution of mesocosms

**Erika M. Hansson** ⬚ *, **Dylan Z. Childs, Andrew P. Beckerman**

School of Biosciences, The University of Sheffield, Sheffield, South Yorkshire, United Kingdom

* emhansson1@sheffield.ac.uk

**Data Availability Statement:** The data presented in this article, along with relevant code, is available from the Zenodo repository (https://doi.org/10.5281/zenodo.6786427).

## Abstract

Microbial experimental evolution allows studying evolutionary dynamics in action and testing theory predictions in the lab. Experimental evolution in chemostats (i.e. continuous flow through cultures) has recently gained increased interest as it allows tighter control of selective pressures compared to static batch cultures, with a growing number of efforts to develop systems that are easier and cheaper to construct. This protocol describes the design and construction of a multiplexed chemostat array (dubbed "mesostats") designed for cultivation of algae in 16 concurrent populations, specifically intended for studying adaptation to herbicides. We also present control data from several experiments run on the system to show replicability, data illustrating the effects of common issues like leaks, contamination and clumps, and outline possible modifications and adaptations of the system for future research.

## Introduction

Microorganisms provide an unparalleled opportunity for the study of evolutionary dynamics due to their combination of short generation time, simple genetics and ability to fit huge population sizes in a small space. The path of evolutionary adaptation can thus be replicated and tightly controlled in real time in the lab, allowing exciting new insights into the mechanisms of adaptive evolution and testing of predictions from theory [1–5].

The most common way of growing microorganisms for experimental evolution is as batch cultures. This involves serial repetition of small cell population subsets being moved to fresh medium and grown to stationary phase before being transferred again to fresh medium to allow new growth and, with time, adaptation. This is a simple, cheap and scalable method, but its drawback is the resulting fluctuating environment as the cells go through "boom and bust"-cycles at every transfer resulting in a complex selective environment [5–7]. As the nutrients in the medium gradually run out, the cells will arrest growth and division while waste products build up. Oxygenation, light levels and pH will also fluctuate with population density. All of this affects cellular metabolism and physiology and subsamples taken from such populations will be growth phase specific, making it difficult to define and isolate the selective pressures

**Funding:** EMH was funded by a University of Sheffield scholarship. The funders had and will not have a role in study design, data collection and analysis, decision to publish, or preparation of the manuscript.

**Competing interests:** The authors have declared that no competing interests exist.

acting on the populations [7]. Furthermore, there is an evolutionary bottleneck at each transfer, where the considerable reduction in population size associated with transfer to the next batch affects the genetic diversity and mutational space available [5, 8–10], giving an increased role to genetic drift in the evolutionary outcome [11].

Chemostats—continuous flow-through, chemically stable cultures where growth medium and treatments are fed into the fixed-volume populations at a constant rate—solve these issues as the specific growth rate of the population at steady state is matched to the dilution rate [12, 13]. The populations are maintained in exponential growth and constant mixing ensures a homogeneous environment, allowing precise control of the relevant selective pressures compared to the complex dynamics present in batch cultures [7]. This unique opportunity for experimental manipulation offers a high-throughput chance to pick apart evolution in action and, as a result, chemostats have recently seen a renaissance in experimental evolution and systems biology as new technological advancements make them easier to maintain than ever before (reviewed in [14, 15]). Chemostats also allow following population fluctuations and evolutionary dynamics in response to experimental treatments in the long term, where equilibria and population cycles including several species and strains can be described as a function of the flow rate (e.g. [16–20]). Multiplexed arrays, where the dilution rate is set by a single pump, and medium sources can be shared, further minimise variation between population chambers [21–27].

Here we describe a multiplexed small-scale DIY chemostat array system (dubbed "mesostats") adapted from the ministat array developed by Miller *et al.* [22] to suit experimental evolution of algae, in contrast to the so far described designs specifically intended for yeast [21, 22, 26] and bacterial cultures [23, 27]. Our system uses common algal model species *Chlamydomonas reinhardtii*, with the specific goal to use it as a herbicide resistance evolution model. *C. reinhardtii* is an established model species for herbicide resistance evolution [8, 28–30] and molecular analysis of herbicide resistance mutations [31–33], but all studies to date have used batch cultures. We present the full protocol for assembling and maintaining a 16-chamber mesostat array by a single person as well as control data illustrating the ability of the system to track trends and variability in the abundance of organisms among replicates. We also present pilot data illustrating the ability to use the mesostats to evolve resistance in *C. reinhardtii* to growth inhibiting herbicide glyphosate. Furthermore, we have included data from this system illustrating the signal of common problems like leaks, contamination and cell clumping, showing how to distinguish it from biological variation as well as how to prevent and address these problems if they occur. We also outline the ways in which this system could be further modified and avenues of future research.

## Methods and materials

The protocol described in this peer-reviewed article is published on protocols.io, https://dx.doi.org/10.17504/protocols.io.6qpvr6q1bvmk/v1 and is included for printing as S1 File with this article.

### Experimental design for validation data

**Replicability.** Presented below are control data from four separate experiments using the linked protocol to show replicability. The conditions and relevant differences for these experiments are summarised in Table 1, unless otherwise stated the experimental conditions correspond to those outlined in the protocol. In all of the presented experiments, *Chlamydomonas reinhardtii* strain Sager's CC-1690 wild-type 21 gr was used, obtained from the *Chlamydomonas* Resource Centre (University of Minnesota, St Paul, MN, USA) core collection. Two

**Table 1. Summary of experimental conditions and properties of data used in Figs 1, 3 and 4.**

| Experiment | n | Dilution rate | Time window | Approximate starting concentration |
|---|---|---|---|---|
| A | 16 (day 1–10) then 4 | 0.3/day | Days 1–20 | 50 000 cells/ml |
| B | 16 (day 1–7) then 4 | 0.3/day | Days 1–28 | 30 000 cells/ml |
| C | 7 (day 3–11) then 2 | 0.15/day | Days 3–35 | 50 000 cells/ml |
| D | 16 (day 5–16) then 6 | 0.15/day | Days 5–76 | 30 000 cells/ml |

different dilution rates were used in the experiments: 0.3/day and 0.15/day. The former was based on the dilution rates used in previous experiments using chemostat populations of similar species that this system was designed for (e.g. [17, 18]), the latter was used as an alternative lower rate to decrease the consumption of growth medium as well as wear and tear on the pump tubing.

**Applicability to experimental herbicide resistance evolution.** Six mesostat chambers in experiment C were allowed a week to reach steady state before the glyphosate treatment was introduced. Shock injections of 38 ml were performed as described in the protocol bringing two chambers each to concentrations of 0 mg/L (controls), 100 mg/L and 150 mg/L glyphosate (analytical standard, PESTANAL®). Both of the chosen glyphosate concentrations are above the minimum inhibitory concentration for *C. reinhardtii* of 97.5 mg/L [30].

**Common problems.** We have provided data from three common problems that present with this type of system: a leak, contamination and algal clumping, all from experiment D. These were spontaneous events and the data presented here aims to show how to identify their signal in the population density data and distinguish it from normal variation among populations. The leak in this example resulted in elevated dilution of a single chamber for roughly four hours due to a clamp securing the pump tubing cassette coming undone. In the case of the contamination event, all of the presented six chambers had been disconnected from the array six days before bacterial contamination was observed under the microscope in four chambers, with the remaining two unaffected by the contamination event. The clumping phenotype was not receiving control medium but presented in a population undergoing treatment with a sublethal dose of glyphosate.

## Sample processing

Population density was in all cases determined through flow cytometry (Beckman Coulter CytoFLEX), using CytExpert (Beckman Coulter) to gate and count events detected in the PerCP-A channel (Excitation: 488nm, Emission: 690/50 BP). This channel is used to detect chlorophyll *a* and represents a robust method for estimating algal density [34] which was further validated against manual haemocytometer counts for this system.

## Data handling

All statistical analyses were carried out in R (version 4.0.5, [35]), using the `lme4` package [36] to fit a linear mixed effects model with log-transformed population density as the response, dilution rate and experiment as fixed effects, and day and chamber as random effects with varying intercepts. The significance of the fixed effects was tested using the Anova() function from the `car` package [37] and confirmed through parametric bootstrapping using the `pbkrtest` package [38].

The slope of population density decline was estimated between days 6–16 with the package `emmeans` [39] after fitting a linear mixed effects model with the log-transformed population

density as the response, treatment and day as fixed effects as well as day and chamber as random effects with varying intercepts.

## Results

### Replicability

Across the experiments presented here, there is no difference between the mean population densities after steady state has been reached ($\chi^2$ = 2.1, DF = 3, p = 0.6, Fig 1). Furthermore, there was no difference in steady state population density whether the dilution rate is 0.3 or 0.15/day ($\chi^2$ = 0.4, DF = 1 p = 0.5). The length of the establishment batch phase before steady state is reached will differ depending on the conditions and inoculate density. The dynamics during this phase has the potential to affect the makeup of the population and thus later dynamics, and it is thus advisable to let the cultures reach steady state before introducing treatments. However, all experiments presented here reached steady state within the first week and it was maintainable for several weeks thereafter.

The level of variation observed in this data set is normal for this type of system [16–18] and can be divided into among population variation and day-to-day variation. Among population variation is primarily caused by the biology of the system as these are separate, genetically heterogeneous populations on separate evolutionary trajectories. Day-to-day variation is however at least partly caused by limitations in sample processing. Both are discussed in more detail in the Discussion section, as well as how to reduce or circumvent the latter in particular.

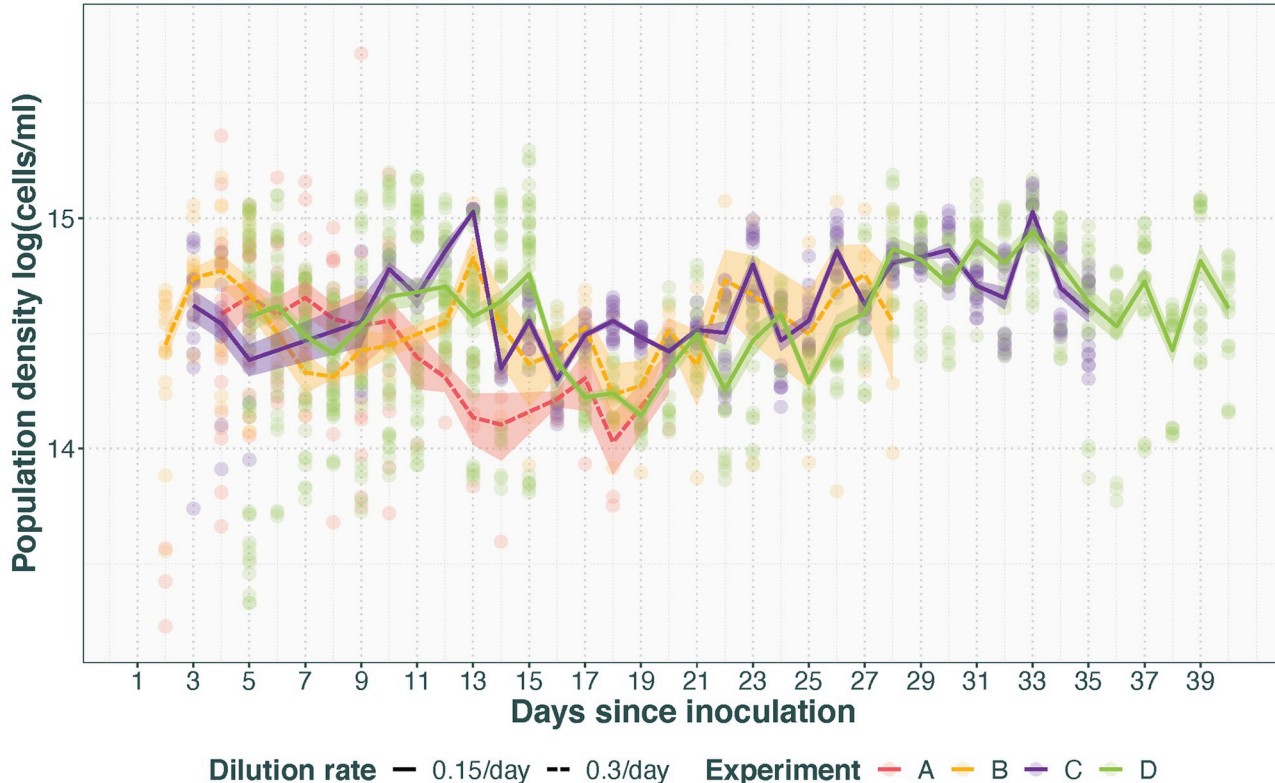

**Fig 1. Population density with time in four separate runs of the mesostat system.** Transparent points represent technical replicates and opaque lines with standard error represent average across populations for experiment. Experiments A and B had a dilution rate of 0.3/day (dashed line), whereas experiments C and D had a dilution rate of 0.15/day (solid line). Note that all have runs have a brief and rapid decline in population density between day 11 and 16. This corresponds to an injection of additional medium as part of the experiment the data is from.

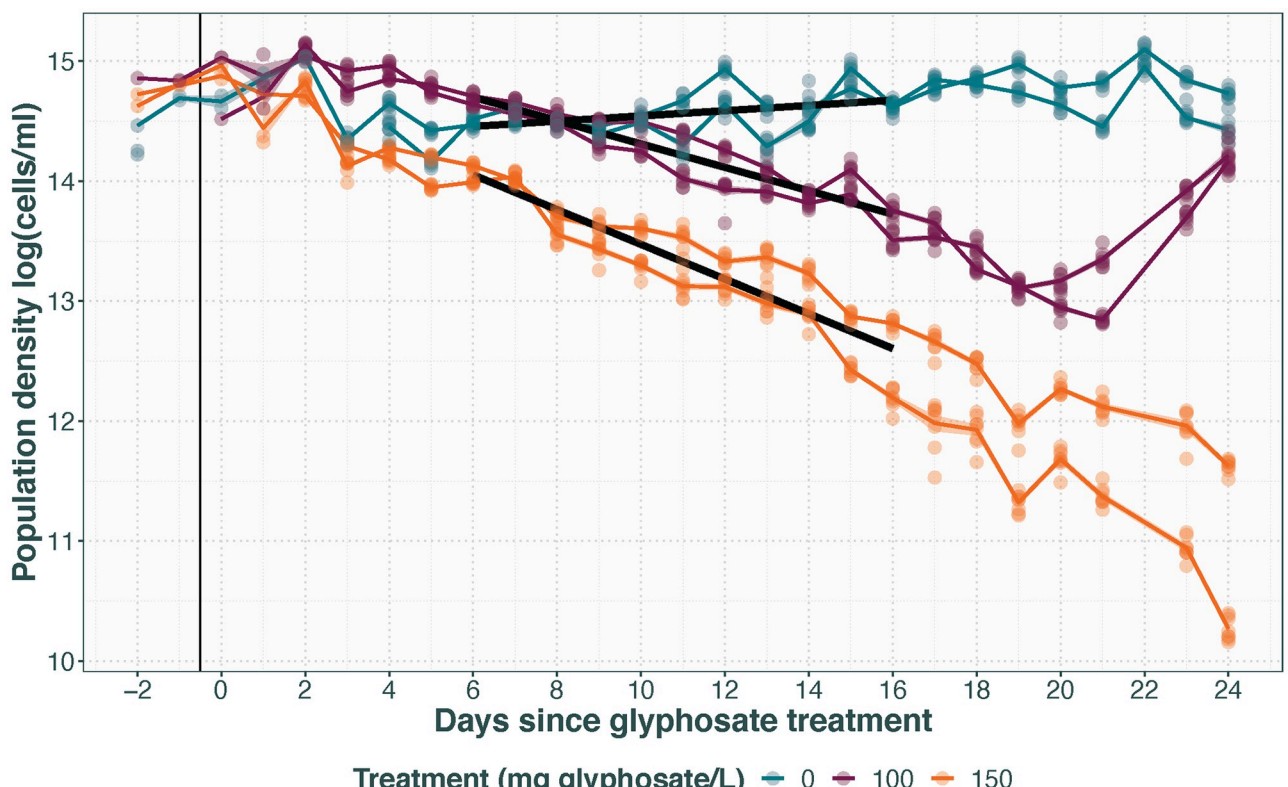

**Fig 2. Population density with time in populations receiving 0, 100 or 150 mg/L glyphosate.** Transparent points represent technical replicates, with opaque lines for population average with standard error transparent ribbon. Thick black lines represent the fitted linear model and the thin black vertical line shows start of the treatment.

## Applicability to experimental herbicide resistance evolution

Fig 2 shows the population densities of the four glyphosate treated populations and two control populations for 24 days following glyphosate treatment introduction. The glyphosate treated chambers exhibit population decline at a rate approximate to (150 mg/L, slope = -0.14, SE = 0.006) or below (100 mg/L, slope = -0.098, SE = 0.006) the dilution rate of 0.15/day. In the same timespan, the control populations exhibit an overall slight increase in population density (slope = 0.022, SE = 0.006), possibly reflecting adaptation to the mesostat environment. The onset of the population decline appears to be immediate for the 150 mg/L glyphosate treatment, whereas it occurs roughly 5 days after the glyphosate injection for the 100 mg/L glyphosate treatment. This is likely due to the 100 mg/L glyphosate treatment being just on the cusp of the minimum inhibitory concentration, enabling the populations to maintain growth for a short while before the herbicidal action is apparent. After 15 and 18 days respectively of population density decline, the 100 mg/L populations increase in cell density again, suggesting the populations have evolved resistance to the glyphosate, whereas the 150 mg/L populations never show evidence of resistance.

## Common problems

**Leaks.** Fig 3 shows the population density in chamber F after a major leak causing over-dilution. Compared to the expected among population and within-population day-to-day variation observed in the chambers that did not experience a leak, three crucial differences

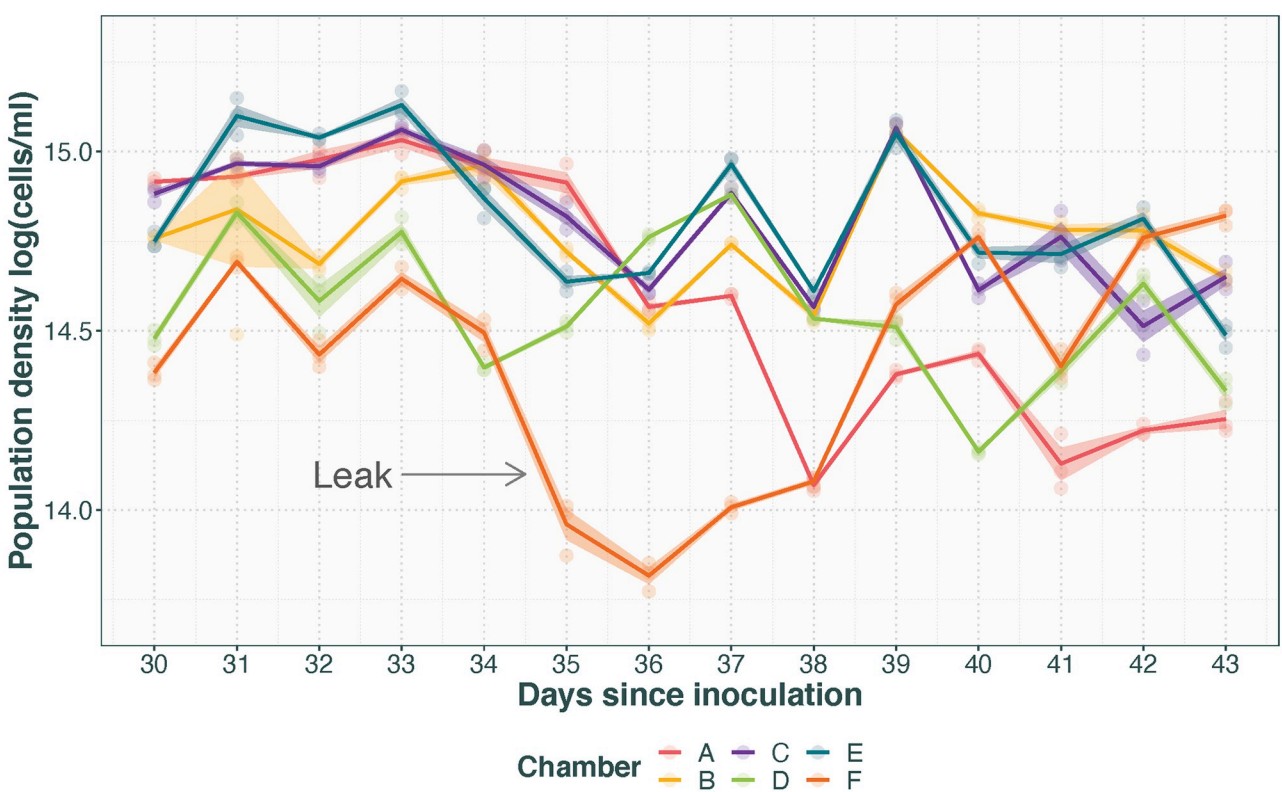

**Fig 3. Population density with time after a major leak.** Transparent points represent technical replicates with opaque lines for population average with standard error. The leak caused overdilution of chamber F between days 34 and 35 (indicated by arrow), compared to the unaffected chambers A–E.

together make this the characteristic signal of over-dilution: 1) While similarly large day-to-day fluctuations in the measured density occur in the presented data set, day effects present across chambers. The rapid reduction in population density for chamber F between days 34 and 35 is only apparent in that chamber, whereas a similar reduction between days 37 and 38 is seen in all of chambers A–E. 2) The reduction in population density in chamber F results in a lower population density than otherwise observed in the data set (by roughly $3 \times 10^5$ cells/ml). 3) The reduced population density is observed in chamber F for several days after day 35, rather than recovering by the next day like seen for chambers A–E after day 38.

**Contamination.** Fig 4 shows the gradual population density decline in four chambers where bacterial contamination was observed under the microscope compared to two chambers that were unaffected by the contamination event. While the average population density of the contaminated chambers starts to trend lower a few days after the contamination event, the full effect on the population density is not clear until several days after the contamination had been observed under the microscope. Furthermore, while there is considerable variation among all populations, the signal of contamination in the data is clearly distinguished from the expected among population variation and day-to-day variation by the fact that it is a consistent, long-term population-density decline without recovery 12 days after the contamination event.

**Clumping.** Fig 5 shows flow cytometry population density estimates from a population exhibiting a clumping phenotype compared to non-clumping populations undergoing the same treatment. The data signal here is an artefact of the limitations of the instrument being

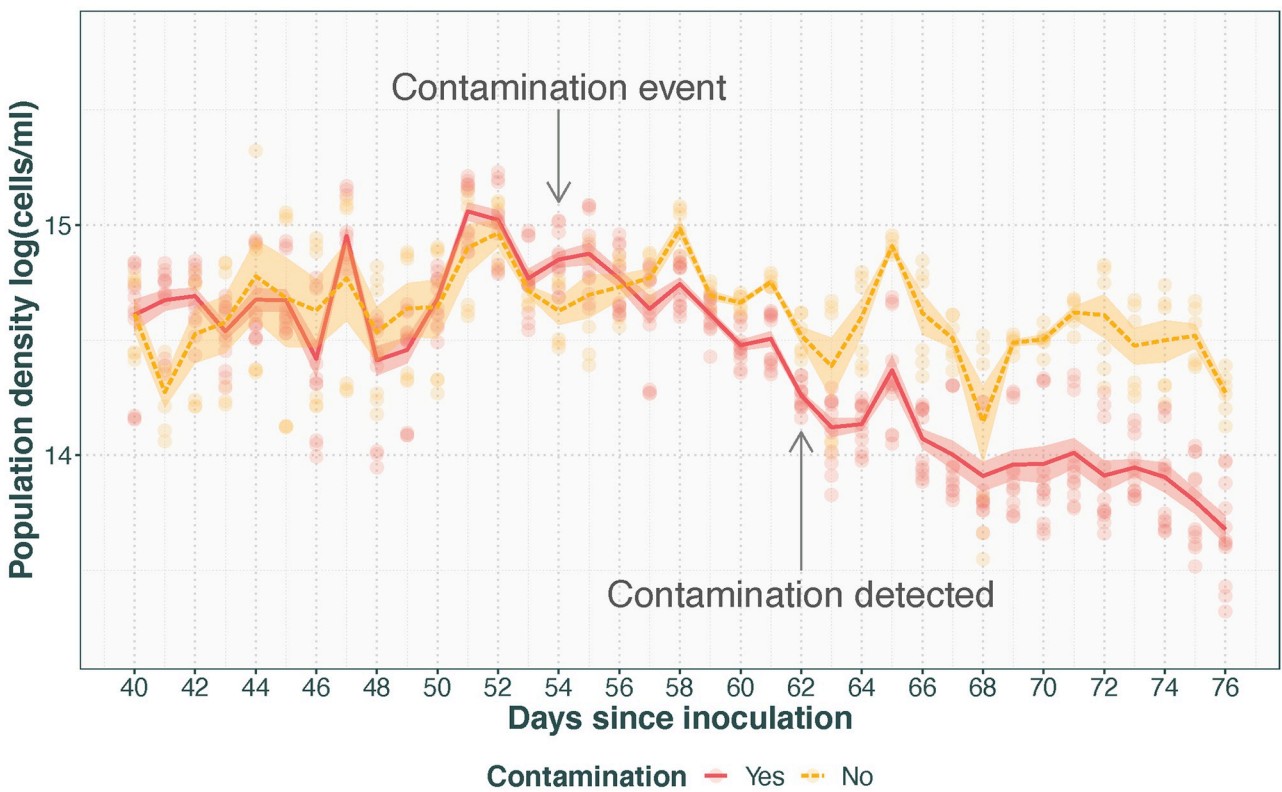

**Fig 4. Population density with time after a contamination event.** Transparent points represent technical replicates with opaque lines with standard error for average of contaminated (solid line) vs. non-contaminated (dashed line) populations. Contamination is likely to have entered the system at day 54 (indicated by arrow), and bacterial contamination was found in 4 out of 6 chambers on day 62 (indicated by arrow).

unable to accurately distinguish individual cells within aggregates, resulting in huge fluctuations in estimated cell density considerably larger than and out of step with the day-to-day variation observed in the other populations.

## Discussion

Chemostats offer a number of advantages over batch cultures for long-term experimental evolution research. Precise control of selective pressures in a chemically constant environment without evolutionary bottlenecks along with a link between growth rate and dilution rate constitute a useful conceptual framework for modelling evolutionary adaptation and population dynamics. This system adds to the small, but growing, number of efforts to produce simple but scalable, multiplexed DIY chemostats from cheaper materials that are possible to build and maintain by a single person [21–27], and is the first of its kind for experimental evolution of algae, specifically the evolution of herbicide resistance in model species *C. reinhardtii*. There are three substantive changes from the Miller *et al.* ministats [22], one system specific and two generic changes to suit experimental evolution with continuous sample extraction. Firstly the system was adapted to suit the study species *C. reinhardtii*, including light and a lower dilution rate, which distinguishes the system from previous DIY chemostat arrays developed for maintenance of yeast [21, 22, 26] and bacterial cultures [23, 27]. Secondly, a needle and syringe system was added to facilitate easy, sterile access to the culture for the removal of samples. This allows sampling from the middle of the active culture rather than relying on the overflow. The efflux only samples from the top and the overflow chamber constitutes a wholly different

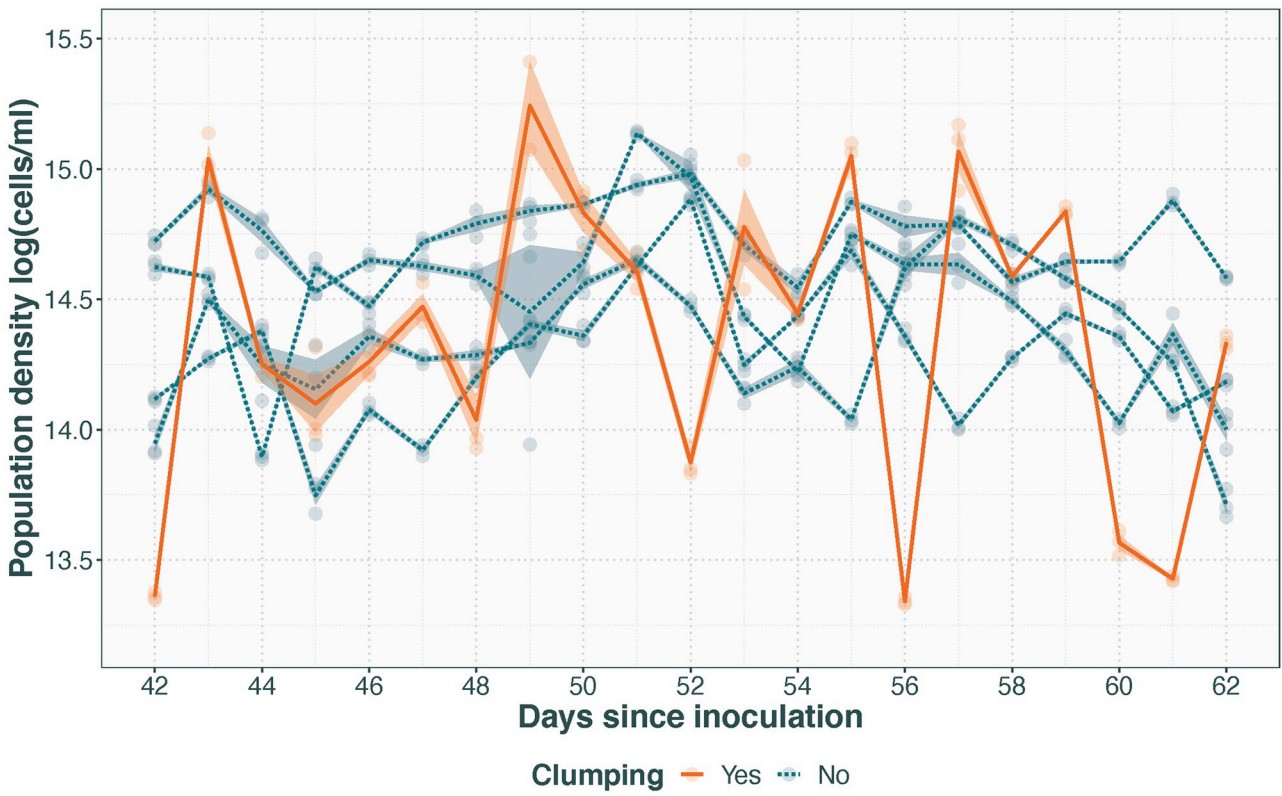

**Fig 5. Population density with time in a population exhibiting a clumping phenotype.** Clumping population shown with solid line, compared to four other populations receiving the same treatment that did not exhibit clumping in dashed line. Transparent points represent technical replicates, with the lines for population averages with standard error.

environment without continuous dilution, build-up of waste products and increased evaporation, making them unrepresentative samples of the chamber populations. Furthermore this simplifies addition of cells or treatment compounds directly to the chambers, eliminating the risk of contamination that comes with disconnecting the medium influx or efflux channels. While sampling ports have been described before (e.g. [23, 25]) our simplification and combination with syringe extraction allows manual sampling with minimal contamination. The third change is an increase in the chamber volume to allow larger population sizes and possible future introduction of several trophic levels. Furthermore, this increases increases the amount of sample that can be extracted on a regular basis, extending the possibilities for the types of assays that can be performed to characterise evolution in action, as most of the previous DIY chemostat arrays have been limited by their small working sizes [22, 23, 25]. Lastly, there were several changes to specific materials to lower the overall costs.

## Sources of variation and how to minimise it

The data presented here illustrates the expected variation between cultures and how to identify the signal of equipment failure, such as a leak, or contamination. We also demonstrate that the system can be used to evolve resistance to growth inhibiting herbicide glyphosate, and that the signal of herbicidal action is apparent as a population density decline, followed by an increase after the population has evolved resistance. The herbicidal effect is clearly distinguishable from the expected variation under control conditions, and given enough time, the resistant population is expected to settle at a new steady state.

The variation among replicate populations observed here is normal [16–18] and expected as they constitute separate, genetically heterogeneous populations on separate evolutionary trajectories. Even when using a single founder population, the genetic bottleneck caused by splitting it between populations as well as the dynamics during the establishment phase of batch-like growth dynamics [6] will result in similar but distinct populations by the time they reach steady state. Effort should be made to ensure that all chambers receive the same levels of light and aeration as well as consistent dilution with the same medium, and starting variation could be eliminated through starting with clone populations at a high enough concentration to effectively avoid the establishment phase. However, the among population variation is generally of scientific interest to experimental evolution studies and should be investigated rather than eliminated.

Conversely, while day-to-day variation within a population is also normal for this type of system, it is also partly caused by limitations to the sampling protocol. The data presented here was obtained from measurements performed on living cells that had the opportunity to grow and divide between sample extraction from the mesostat chambers and sampling processing. While this is an unavoidable source of variation, it can however be reduced by minimising the time that passes and working in a controlled temperature environment. If the experimental design allows, the cells can be immobilised by using e.g. Lugol's solution before counting with flow- or haemocytometry. It is also possible to control for this variation by including sampling day as a source of error in statistical models applied to the data.

The among population and day-to-day within population variation are however both clearly distinguishable from the data signal of common faults like leaks, contamination and clumping. While these faults are likely to be detected before they become apparent in the population density data, leaks causing significant over-dilution are apparent within a few hours while clumping and contamination can be observed under a microscope, it is important to understand how they affect the data so that an informed decision can be made on how to handle it. While the population density is always expected to quickly return to steady state after over-dilution, the increased flushing out of cells constitutes an evolutionary bottleneck and the changed growth conditions may affect other traits of the population not visible in the population density data and data collected subsequent to a major leak should thus be treated with caution. The leak presented here was caused by equipment failure resulting in over-dilution, but smaller leaks often occur as the pump tubing wears out with long term use, which can lead to under-dilution of the connected chambers. Both are best prevented by regular inspection of the pump parts for irregularities.

Bacterial contamination is another common risk in long-running continuous cultures [15], and is best prevented by working in a sterile environment and minimising the points at which contamination can enter the system. The main contamination risk presents when disconnecting any part of the array, such as when switching medium containers, or when extracting samples, and particular care should be taken to keep the connecting parts sterilised during. The example presented here is the only instance of contamination observed across eight separate experiments each lasting more than a month and happened when the chambers were disconnected from the array for a longer period of time and removed from the sterile environment. Even so, only four out of six chambers showed evidence of contamination under the microscope 12 days after the contamination event, despite all of the chambers in question sharing a medium source. This suggests that the system is robust in terms of contamination not spreading between the chambers. While regular microscopy inspection of cell samples for contamination is recommended, this can be laborious with a large number of replicates and the characteristic population density decline provides another opportunity to detect and isolate the problem.

Lastly, chemostat populations being under a selective pressure to evolve phenotypes reducing their risk of being flushed out is an often cited issue with the method [7, 15], presenting as adhesion to the chamber walls and cell flocculations. While this phenomenon has as of yet never been observed under control conditions with this protocol, there was one instance of cells exhibiting a clumping phenotype while under treatment with a sublethal dose of glyphosate, making it possible it was a response to the treatment rather than the mesostat environment. In *C. reinhardtii* there are two distinct types of clumping: cell aggregations of separate cells and palmelloid colonies that share a cell wall [40–42]. Both have been found to be an induced response as well as heritable [40, 41, 43–45], meaning that once they become common in a population they may be hard to get rid of [40]. Palmelloids are small enough that they will not cause blockages, but due to the shared cell wall they cannot be disassociated through bubbling or by vortexing a sample. Cell aggregations can be considerably larger, however they are also possible to break apart through vortexing, and vigorous bubbling of the cultures often prevents their formation [7]. How much of a problem clumping is depends on the experiment, i.e. it becomes a problem if it hinders sample processing and when it is thought to be an artefact of the chemostat environment rather than in response to the applied treatment. For population density measurements by flow cytometry as presented here, clumping considerably reduces the accuracy of the measurements as each clump is counted as a single particle, increasing day-to-day variation. In this case, manual haemocytometry could give a better estimate but this is considerably more laborious.

## Other possible issues and limitations

Despite the many advantages of chemostat cultures, there are limitations to their application and caveats to how the data may be interpreted. While the system described in this protocol was explicitly designed to be maintainable by one person as well as cheaper than the Miller *et al.* ministats [22] it is based on by choosing alternative materials and using parts not purpose bought for this experiment, it is still considerably more expensive than batch cultures. While it is theoretically possible to run very large cultures indefinitely, the cost of the medium or treatment components will limit the lifespan of the experiment as they will be consumed faster than in a batch culture design. One way to conserve medium and treatment components is to lower the dilution rate, which in the experiments presented here had no effect on population density in the chambers. However, this changes the selection pressures experienced by the populations as well as their doubling rate [15]. The logistics of the system and any cost saving measures must therefore be carefully balanced against the resulting biology, taking into account the desired selective pressure, cell cycle stage and generation time.

This design introduced sampling needles to allow sampling directly from the culture as an alternative to sampling from the overflow chamber, as the environment therein will be different from the culture chamber, or redirecting the overflow, as the low flow rate made sampling a slow process and the high temperature caused high levels of evaporation. However, sampling directly from the culture does perturb the steady state and change the dynamics within the chamber by temporarily reducing the culture volume and thus pausing dilution [46]. The frequency and volume of samples should thus be carefully considered against the disruption they may cause.

Another potential problem involves insufficient aeration or efflux blockages causing over- or under-pressure in the chambers. Provided the air supply is sufficient, the most common reason for low or uneven bubbling is blocked air filters, usually because they have become damp. If the air filters frequently become damp, the ambient humidity may be too high. Not enough bubbling may cause sedimentation and stratification of the culture, as well as selection

for phenotypes that sink so they avoid being flushed out, or it may instead cause the culture to rise through the aeration needle instead of through the efflux needle, changing the effective dilution rate. Clogging of the media line is uncommon, but can occur if not properly sterilised and contamination is allowed to grow. This is often apparent as a reduction in flow into and out of the affected chambers. The daily maintenance part of the protocol outlines how to spot and address these problems.

Lastly, in terms of studying the dynamics of adaptive evolution, chemostat systems are highly specific environments. When transplanted out, chemostat populations are often found to grow poorly in their ancestral environment compared to the ancestral strain [15, 47], as they have had intense selection on a specific part of the growth cycle in an environment of constant dilution that is not reflective of natural populations. However, this is also part of their usefulness and beauty, by keeping the adaptive environment as simple and specific as possible, we can isolate fitness effects and allow fine-tuned investigation of their mechanics and dynamics.

## How to improve or modify

Several further modifications are possible for this system. A light table that does not transfer heat to the cultures would allow the internal culture temperature to be set solely by the ambient temperature in the controlled temperature room while maintaining low evaporation. As the chamber lids are relatively large, sensors to monitor e.g. pH or $CO_2$ levels could also be fitted through additional ports (see [25]).

While the pump and pump-tubing are integral to the design and also the most expensive parts, all other parts could be easily substituted depending on availability or cost constraints. The materials list provided in the protocol can be used as a guide for the dimensions and properties of the part, but primarily aims to illustrate how this type of system can be built from parts already found in most wet labs rather than buying a pre-made set. Any water-tight, sterilisable container can be used for culture chambers if suitable lids can be manufactured, such as falcon tubes [23] or commonly available lab glassware [25]. The controlled temperature room can be replaced with water baths (note however that this requires mounting the lights up on the sides of the water baths), and portable aquarium pumps can be used instead of building infrastructure, increasing flexibility in where the system can be housed.

The light system here is rudimentary but sufficient for *C. reinhardtii* growth [40], using white light LED strip lights mounted around the chambers along with a DIY light box also consisting of white light LED strip lights and a semi-transparent plastic top to diffuse the light. The light box is not necessary, but convenient for maximising light from all angles. Under control conditions 24h light was used, but it is possible to fit a timer to the outlet connecting the lights to instead provide a diurnal light cycle. Coloured semi-transparent plastic could be used to provide light only from a specific part of the light spectrum, but it would also be possible to mount specialist lighting around the mesostats if tuning for a specific photosynthetic organism or experiment is desired.

## Future research opportunities

We have used this system for experimental evolution of herbicide resistance in algae by adding glyphosate as a shock injection and then continuously through the growth medium, however, this setup is also easily adaptable depending on the research question. The herbicide treatment could also be applied gradually through the medium or through series of shock injections in a ratchet protocol [28] and investigate to what level the resistance can be pushed and at what speed. The dilution rate and thus the cell growth rate is set by the pump speed, tubing

thickness and culture volume, so running chambers with different dilution rates simultaneously would be possible with different pump tubing thicknesses, multiple pumps or multiple culture volumes, depending on the range required. Furthermore, the use of multiple light tables with opaque partitions between cultures would allow testing for an interaction with light level, or the chambers could be kept in water baths at different temperatures to determine the effect of temperature.

In addition to testing the effect of abiotic factors such as temperature or light, or manipulating the specific cell cycle stage of the population, a particularly interesting future application would be to use the system to ask focused questions about eco-evolutionary dynamics. In particular, introducing several trophic levels in the culture chambers to study the ecosystem and food web effects of herbicides and evolving herbicide resistance. The predator-prey cycles of rotifer *Brachionus calyciflorus* and *C. reinhardtii* as well as *Chlorella vulgaris* have been successfully studied and modelled using chemostat environments (e.g. [17, 18]) and our setup allows simplified simultaneous replication for this type of system that can be maintained by one person. Competition could also be introduced to the system through using multiple algal strains and monitoring their frequencies or through expanding the culture ecosystem to include other algal species or bacteria [48].

## Supporting information

**S1 File. Step-by-step protocol, also available on protocols.io.**
(PDF)

**S1 Fig. Simplified schematic of the mesostat system.** The mesostat system includes medium container(s), a pump, culture chamber(s) with sampling needle, overflow bottle(s), a gas washing bottle along with the medium and air influx lines and the culture efflux line.
(TIF)

**S2 Fig. Photographs of the mesostat system.** A) The complete setup just after inoculation with algae, running an experiment with six levels of treatments applied through the media lines. B) Close-up of medium siphon through medium container lid. C) Close-up of connection between pump tubing and silicone tubing used throughout array. D) Close-up of culture chambers rubber bung with the four hypodermic needles, capped sampling needle to the left in foreground, steel efflux needle to the right in foreground, steel aeration needle in the middle, and pink plastic medium influx needle in the background. E)The overflow chamber (left) and the culture chamber at steady state (right) with the efflux line running between them.
(TIF)

**S3 Fig. Assembly schematic; the media lines.**
(TIF)

**S4 Fig. Assembly schematic; the chambers.**
(TIF)

**S5 Fig. Assembly schematic; the aeration line.**
(TIF)

## Acknowledgments

Special thanks to Allison Blake and Lynsey Gregory for technical support in the lab.

## Author Contributions

**Conceptualization:** Erika M. Hansson, Dylan Z. Childs, Andrew P. Beckerman.

**Data curation:** Erika M. Hansson.

**Formal analysis:** Erika M. Hansson.

**Funding acquisition:** Dylan Z. Childs, Andrew P. Beckerman.

**Investigation:** Erika M. Hansson.

**Methodology:** Erika M. Hansson, Dylan Z. Childs, Andrew P. Beckerman.

**Resources:** Dylan Z. Childs, Andrew P. Beckerman.

**Supervision:** Dylan Z. Childs, Andrew P. Beckerman.

**Validation:** Erika M. Hansson.

**Visualization:** Erika M. Hansson.

**Writing – original draft:** Erika M. Hansson.

**Writing – review & editing:** Dylan Z. Childs, Andrew P. Beckerman.

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
