## [Decision Letter · Decision Letter 0]

26 May 2022

PONE-D-22-12361Mesostats – A multiplexed, low-cost, do-it-yourself continuous culturing system for experimental evolution of mesocosmsPLOS ONE Dear Dr. Hansson,

Thank you for submitting your manuscript to PLOS ONE. After careful consideration, we feel that it has merit but does not fully meet PLOS ONE’s publication criteria as it currently stands. Therefore, we invite you to submit a revised version of the manuscript that addresses the points raised during the review process. Please do not consider resubmitting your manuscript in case the points raised during the review process have not been exhaustively addressed.

Main points:

Discrepancy with the publication criteria for Lab Protocol type of article. In addition to the general PlosOne criteria for publication, the manuscripts should provide clear evidence that the protocol works and that it is not presenting describing routine methods, or extensions or modifications of routine methods. In this context, a clear outline of the novelty of this system along with a more detailed idea of what was done with this system are required for publication.Please, ignore the request of providing a detailed description of methods by Reviewer 1Take into carful consideration all points raised by Reviewers, particularly attention should be paid to request related to validation and reproducibility.Align manuscript presentation with journal requirements for Lab Protocol as presented in the Template available on the dedicated link (https://journals.plos.org/plosone/s/submission-guidelines#loc-lab-protocols).

We look forward to receiving your revised manuscript.

Kind regards,

Maya Dimova Lambreva, Ph.D.

Academic Editor

PLOS ONE

Journal Requirements:

2. To comply with PLOS ONE submissions requirements, please provide the Protocols.io DOI in the Methods section of the manuscript using this format: “The protocol described in this peer-reviewed article is published on protocols.io, https://dx.doi.org/10.17504/protocols.io[........] and is included for printing as supporting information file 1 with this article.” Please also provide the Protocols.io DOI in the “Protocol DOI” field of the submission form (via “Edit Submission”). For more information, please see our submission guidelines:  https://journals.plos.org/plosone/s/submission-guidelines#loc-guidelines-for-specific-study-types

3. We note you have not provided a Protocol.io PDF version of your protocol. As noted in our submission requirements, please upload a Protocol.io PDF version of your protocol as a Supporting Information file and name the file ‘S1 file’. Please update your Supporting Information Captions if necessary. If you have not yet uploaded your protocol to Protocols.io you are welcome to use the Protocols.io customer service code ‘PLOS2021.’ When using this customer code while submitting to Protocols.io, please make reference to your PLOS ONE submission, including your PLOS ONE manuscript number. With this customer code, Protocols.io editorial staff will import and format your protocol at no charge. For more information, please see our submission guidelines:  https://journals.plos.org/plosone/s/submission-guidelines#loc-guidelines-for-specific-study-types

"EMH was supported by a scholarship from The University of Sheffield. Special thanks to Allison Blake and Lynsey Gregory for technical support in the lab. "

"EMH was funded by a University of Sheffield scholarship. The funders had and will not have a role in study design, data collection and analysis, decision to publish, or preparation of the manuscript."

5. We note you have included a table to which you do not refer in the text of your manuscript. Please ensure that you refer to Table 1 in your text; if accepted, production will need this reference to link the reader to the Table.

Reviewers' comments:

Reviewer's Responses to Questions

**Comments to the Author**

1. Does the manuscript report a protocol which is of utility to the research community and adds value to the published literature?

Reviewer #1: No

Reviewer #2: Yes

2. Has the protocol been described in sufficient detail?

Descriptions of methods and reagents contained in the step-by-step protocol should be reported in sufficient detail for another researcher to reproduce all experiments and analyses. The protocol should describe the appropriate controls, sample sizes and replication needed to ensure that the data are robust and reproducible.

Reviewer #1: No

Reviewer #2: Yes

3. Does the protocol describe a validated method?

Reviewer #1: No

Reviewer #2: Yes

4. If the manuscript contains new data, have the authors made this data fully available?

Reviewer #1: N/A

Reviewer #2: Yes

**5. Is the article presented in an intelligible fashion and written in standard English?**

Reviewer #1: Yes

Reviewer #2: Yes

6. Review Comments to the Author

Reviewer #1: Major comments:

Which cultures/associations have been used? What were their initial parameters (cell densities, DW, etc.)?? Mere including another paper as a supplementary file with the method description is not enough—at least, a brief recapitulation of the key methods used should be provided.

The section ”Representative results and observations” is actually a mix of Methods and Results.

The authors underline the economical aspects of their system but never present cost estimation/breakdown.

As a result of overall lack of structure, it is difficult to understand which scientific goal was set and achieved in this study, and the whole manuscript is more like a technical note with a limited novelty than a scientific paper.

The authors can be advised to set forth a scientifically relevant problem and solve it with the proposed mesostat(s) to demonstrate its applicability.

Minor comments:

Introduction: Here I describe a… — please rephrase.

The authors mention “…this thesis..” in the section called “Limitations”, but this is an article, not a thesis, right?

Reviewer #2: The manuscript describes a development of a low cost mesoscale continuous culturing system, which could be applied to investigate adaptive evolution of microalgae. The method has potential utility for research and describes some new developments, and the methods and materials are described in sufficient details for reproducibility. However some improvements are necessary for method validation and other points as indicated below.

line 77-78, replicability: it is not clear why only these dilution rates (0.15 and 0.3 /day) were applied. Is there a range of dilution rates, based on which the ideal dilution rate (i.e. to achieve steady state) could be established? Did the authors test some 'extreme' range of dilution rates to test the plasticity of the setup?

Figure 2: the results of this experiment are not entirely clear, because there seems to be quite a large variations in treatments A-E as well, which did not undergo leak test. Conditions A-E need to be better explained.

Figure 3: There is some variability of population density in the control, non-contaminated cultures as well, so these results cannot be judged with certainty. Authors refer to 'likely' source of contamination, could not this be better verified with deliberately contaminating some of the cultures, e.g. by inoculating some bacteria?

Besides the common problems investigated, are there any potential problems with the introduced mesoscale chemostat system? For example changes in system pressure, aeration, problems with cell clumping, clogging of the tubing system and needles, etc. The impact of these potential problems should be mentioned and possible solutions offered to circumvent these issues.

line 39: perhaps plural pronoun should be used throughout, as there are multiple authors of the work.

line 118-119 - if the work was the basis of a thesis, add the relevant reference of the thesis.

line 128-129: 'the system must therefore be carefully balanced against the resulting biology' -this is unclear, please clarify/define this more precisely.

line 133-135: are there pilot experiment performed on the potential effects of sampling frequency on population density and potential disruption of the system?

line 148-150: In a high performance chemostat system, these sensors are indeed important, therefore the authors need to demonstrate the feasibility of introducing these sensors in their system.

The light system lacks some details. What was the light source applied? Could the authors provide the emission spectrum of the lamp? Is it possible to provide diurnal light cycles? Is it possible to change the spectral composition of the light source? So could it be tuned for different photosynthetic organisms with different photosynthetic pigment or light harvesting antenna composition?

In the Discussion, authors describe several potential future improvements and research opportunities, but how feasible are these practically for the introduced system? For example, temperature control seems to be unsolved for the current stage of development, and the application of e.g. a water bath may need a significant redesign of the entire system. The feasibility of the proposed applications and future developments should be compared and analyzed with other existing continuous culturing systems.

7. PLOS authors have the option to publish the peer review history of their article (what does this mean?). If published, this will include your full peer review and any attached files.

Reviewer #1: No

Reviewer #2: No

---

## [Author Response · Author response to Decision Letter 0]

2 Jul 2022

We thank the editor and the referees for their constructive comments. We have addressed the substantive concerns that appear to relate to placing our protocol in the wider context of other related methods and ensuring that aspects of validation, clarity, and the reproducibility of the protocol are made more clear.

Below we respond first to the PLOS editorial comments and then below, point by point, to the referees in the PLOS template. All major changes have also been copied into the rebuttal.

PLOS

Main points:

Discrepancy with the publication criteria for Lab Protocol type of article. In addition to the general PlosOne criteria for publication, the manuscripts should provide clear evidence that the protocol works and that it is not presenting describing routine methods, or extensions or modifications of routine methods. In this context, a clear outline of the novelty of this system along with a more detailed idea of what was done with this system are required for publication.

RESPONSE: We thank the editor for this comment. We have enhanced citations of similar work that supports the motivation and context of the work and places our work more effectively in the wider literature, as well as highlight that this protocol is the first description of this type of system adapted for experimental evolution of algae in addition to being used for modelling herbicide resistance evolution. 

Specifically the novelty of the system is addressed in the Introduction and is expanded on in detail in the Discussion and have been pasted below in response to Reviewer #1. 

Furthermore, to make clear what the system was used for, we have modified our presentation of data and enhanced the evidence it provides that the protocol can track trends and variability in the abundance of organisms among replicates. 

Finally, we have also added pilot data demonstrating the applicability of the system to the intended use of experimental herbicide resistance evolution, showing how evolution in action can be followed through population density data and that this effect is clearly distinguishable from among population variation, day-to-day variation or the variation introduced by equipment failures. Details of these changes have been pasted below in response to the relevant comments by Reviewer #2.

Please, ignore the request of providing a detailed description of methods by Reviewer 1

RESPONSE: Thank you.

Take into carful consideration all points raised by Reviewers, particularly attention should be paid to request related to validation and reproducibility.

RESPONSE: Thank you for this advice. We have improved our description of the motivation and context of the work, extended the description and discussion of biological variation of experimental interest vs. variation caused by equipment failure as well as addressed further possible issues and improvements in more detail which are the key points raised by the reviewers. The changes have been pasted below in response to the relevant comments.

Align manuscript presentation with journal requirements for Lab Protocol as presented in the Template available on the dedicated link (https://journals.plos.org/plosone/s/submission-guidelines#loc-lab-protocols).

RESPONSE: Thank you for this advice. We have done this, see responses below:

RESPONSE: Thank you for pointing this out. We have done the following to address this: 

Changed the title of Table 1 to conform to style guidelines.

Changed file names of the supporting information protocol file as well as figures to “S1_File.pdf” and “S1_Fig1.tiff” etc.

Added a Supporting Information section (see pasted below in response to point 6).

Removed the Funding Statement from the manuscript (see below).

Uploaded the protocol to Protocols.io.

2. To comply with PLOS ONE submissions requirements, please provide the Protocols.io DOI in the Methods section of the manuscript using this format: “The protocol described in this peer-reviewed article is published on protocols.io, https://dx.doi.org/10.17504/protocols.io[........] and is included for printing as supporting information file 1 with this article.” Please also provide the Protocols.io DOI in the “Protocol DOI” field of the submission form (via “Edit Submission”). For more information, please see our submission guidelines: https://journals.plos.org/plosone/s/submission-guidelines#loc-guidelines-for-specific-study-types

RESPONSE: Thank you for pointing this out, the Protocols.io DOI has now been included in the manuscript as well as in the relevant field in the submission form.

3. We note you have not provided a Protocol.io PDF version of your protocol. As noted in our submission requirements, please upload a Protocol.io PDF version of your protocol as a Supporting Information file and name the file ‘S1 file’. Please update your Supporting Information Captions if necessary. If you have not yet uploaded your protocol to Protocols.io you are welcome to use the Protocols.io customer service code ‘PLOS2021.’ When using this customer code while submitting to Protocols.io, please make reference to your PLOS ONE submission, including your PLOS ONE manuscript number. With this customer code, Protocols.io editorial staff will import and format your protocol at no charge. For more information, please see our submission guidelines: https://journals.plos.org/plosone/s/submission-guidelines#loc-guidelines-for-specific-study-types

RESPONSE: Thank you for this information, we have now provided a Protocols.io pdf version of the protocol as instructed.

"EMH was supported by a scholarship from The University of Sheffield. Special thanks to Allison Blake and Lynsey Gregory for technical support in the lab. "

"EMH was funded by a University of Sheffield scholarship. The funders had and will not have a role in study design, data collection and analysis, decision to publish, or preparation of the manuscript."

RESPONSE: Thank you for pointing this out. The funding information has been removed from the manuscript. The current Funding Statement is correct and can be kept as is.

5. We note you have included a table to which you do not refer in the text of your manuscript. Please ensure that you refer to Table 1 in your text; if accepted, production will need this reference to link the reader to the Table.

RESPONSE: Thank you for noticing this, the table in question had been mislabelled as a figure which has been amended.

RESPONSE: Thank you for pointing this out. We have now included the following Supporting information section (lines 385–407):

Supporting information

S1 File

Step-by-step protocol, also available on protocols.io.

S1 Fig1

Simplified schematic of the mesostat system. The mesostat system includes medium container(s), a pump, culture chamber(s) with sampling needle, overflow bottle(s), a gas washing bottle along with the medium and air influx lines and the culture efflux line.

S1 Fig2

Photographs of the mesostat system. A) The complete setup just after inoculation with algae, running an experiment with six levels of treatments applied through the media lines. B) Close-up of medium siphon through medium container lid. C) Close-up of connection between pump tubing and silicone tubing used throughout array. D) Close-up of culture chambers rubber bung with the four hypodermic needles, capped sampling needle to the left in foreground, steel efflux needle to the right in foreground, steel aeration needle in the middle, and pink plastic medium influx needle in the background. E)The overflow chamber (left) and the culture chamber at steady state (right) with the efflux line running between them.

S1 Fig3

Assembly schematic; the media lines.

S1 Fig4

Assembly schematic; the chambers.

S1 Fig5

Assembly schematic; the aeration line.

REFEREE COMMENTS

2. Has the protocol been described in sufficient detail?

Descriptions of methods and reagents contained in the step-by-step protocol should be reported in sufficient detail for another researcher to reproduce all experiments and analyses. The protocol should describe the appropriate controls, sample sizes and replication needed to ensure that the data are robust and reproducible.

Reviewer #1: No

Reviewer #2: Yes

3. Does the protocol describe a validated method?

Reviewer #1: No

Reviewer #2: Yes

4. If the manuscript contains new data, have the authors made this data fully available?

Reviewer #1: N/A

Reviewer #2: Yes

5. Is the article presented in an intelligible fashion and written in standard English?

Reviewer #1: Yes

Reviewer #2: Yes

6. Review Comments to the Author

Reviewer #1: Major comments:

Which cultures/associations have been used? What were their initial parameters (cell densities, DW, etc.)??

RESPONSE: Thank you for this comment. We have included the starting concentrations for each experiment in Table 1 as well as added the cultures used to Methods section (see pasted below, from lines 65–68):

In all of the presented experiments, Chlamydomonas reinhardtii strain Sager’s CC-1690 wild-type 21 gr was used, obtained from the Chlamydomonas Resource Centre (University of Minnesota, St Paul, MN, USA) core collection.

Mere including another paper as a supplementary file with the method description is not enough—at least, a brief recapitulation of the key methods used should be provided.

RESPONSE: We have ignored this on advice from the Editor.

The section ”Representative results and observations” is actually a mix of Methods and Results.

RESPONSE: We have moved the parts that constituted methods to a subsection in the Methods (pasted below), conforming to the format used by other papers published by PLOS ONE (e.g. https://doi.org/10.1371/journal.pone.0263071 or https://doi.org/10.1371/journal.pone.0259202), see pasted below (copied from lines 60–110):

Experimental design for validation data

Replicability

Presented below are control data from four separate experiments using the linked protocol to show replicability. The conditions and relevant differences for these experiments are summarised in Table 1, unless otherwise stated the experimental conditions correspond to those outlined in the protocol. In all of the presented experiments, Chlamydomonas reinhardtii strain Sager’s CC-1690 wild-type 21 gr was used, obtained from the Chlamydomonas Resource Centre (University of Minnesota, St Paul, MN, USA) core collection. Two different dilution rates were used in the experiments: 0.3/day and 0.15/day. The former was based on the dilution rates used in previous experiments using chemostat populations of similar species that this system was designed for (e.g. [17, 18]), the latter was used as an alternative lower rate to decrease the consumption of growth medium as well as wear and tear on the pump tubing.

Applicability to experimental herbicide resistance evolution

Six mesostat chambers in experiment C were allowed a week to reach steady state before the glyphosate treatment was introduced. Shock injections of 38 ml were performed as described in the protocol bringing two chambers each to concentrations of 0 mg/L (controls), 100 mg/L and 150 mg/L glyphosate (analytical standard, PESTANAL®). Both of the chosen glyphosate concentrations are above the minimum inhibitory concentration for C. reinhardtii of 97.5 mg/L [30].

Common problems

We have provided data from three common problems that present with this type of system: a leak, contamination and algal clumping, all from experiment D. These were spontaneous events and the data presented here aims to show how to identify their signal in the population density data and distinguish it from normal variation among populations. The leak in this example resulted in elevated dilution of a single chamber for roughly four hours due to a clamp securing the pump tubing cassette coming undone. In the case of the contamination event, all of the presented six chambers had been disconnected from the array six days before bacterial contamination was observed under the microscope in four chambers, with the remaining two unaffected by the contamination event. The clumping phenotype was not receiving control medium but presented in a population undergoing treatment with a sublethal dose of glyphosate.

Sample processing

Population density was in all cases determined through flow cytometry (Beckman Coulter CytoFLEX), using CytExpert (Beckman Coulter) to gate and count events detected in the PerCP-A channel (Excitation: 488nm, Emission: 690/50 BP). This channel is used to detect chlorophyll a and represents a robust method for estimating algal density [34] which was further validated against manual haemocytometer counts for this system.

Data handling

All statistical analyses were carried out in R (version 4.0.5, [35]), using the lme4 package [36] to fit a linear mixed effects model with log-transformed population density as the response, dilution rate and experiment as fixed effects, and day and chamber as random effects with varying intercepts. The significance of the fixed effects was tested using the Anova() function from the car package [37] and confirmed through parametric bootstrapping using the pbkrtest package [38].

The slope of population density decline was estimated between days 6–16 with the package emmeans [39] after fitting a linear mixed effects model with the log-transformed population density as the response, treatment and day as fixed effects as well as day and chamber as random effects with varying intercepts.

The authors underline the economical aspects of their system but never present cost estimation/breakdown.

RESPONSE: We thank the reviewer for this comment — however the point of the paper and described method is to provide an example of how easily this type of system can be built from parts readily found from all major lab material suppliers and parts that are likely to already be present in most wet labs, rather than buying a premade set. 

We have provided a detailed list of components as part of the protocol and, the prices and availability of them is subject to change and to vary internationally. Our emphasis is on that the vast majority of those parts could be substituted for something similar according to need.

To add clarity around this issue, the following sentence has been added to the Materials list in the protocol: 

Other than the pump and pump tubing, all of the pieces are fairly standard pieces found in many wet labs and similar products can be obtained easily from all major scientific suppliers.

Furthermore, the How to Improve or Modify subsection in the Discussion section addressing substitution of parts has been expanded to the following (lines 342–347):

While the pump and pump-tubing are integral to the design and also the most expensive parts, all other parts could be easily substituted depending on availability or cost constraints. The materials list provided in the protocol can be used as a guide for the dimensions and properties of the part, but primarily aims to illustrate how this type of system can be built from parts already found in most wet labs rather than buying a pre-made set.

As a result of overall lack of structure, it is difficult to understand which scientific goal was set and achieved in this study, and the whole manuscript is more like a technical note with a limited novelty than a scientific paper.

RESPONSE: We thank the referee for this comment. We have made every effort to follow the guidelines for a protocol submission which is quite different to a study with a scientific goal. To make the objective and application of the protocol more clear, we have increased the context of the work and the data/examples.

First, with respect to context, our protocol thus contributes to an emerging landscape of protocols and design criteria for experimental evolution, and highlights specifically how to implement this for algae, larger volumes and higher replication for experimental evolution within the eco-evolutionary context with the specific view to apply it to herbicide resistance evolution.

We are aware of and reference and discuss two additional manuscripts that have added evidence for how to specifically expand the Miller et al (2013) platform, initially developed for yeast work. One (Tonoyan et al. 2020. Construction and Validation of A Low-cost, Small-scale, Multiplex Continuous Culturing System for Microorganisms) focuses on developing a cost-effective system for working with bacteria and the other (Ekkers et al. 2020. The Omnistat: A Flexible Continuous-Culture System for Prolonged Experimental Evolution) focuses on a bespoke, highly refined system for microbial communities. The unique and complementary aspects of our system are now expressed more clearly in the introduction and more fully developed in the discussion (see changes pasted below):

Introduction Content:

Lines 39–55: Here we describe a multiplexed small-scale DIY chemostat array system (dubbed “mesostats”) adapted from the ministat array developed by Miller et al. [22] to suit experimental evolution of algae, in contrast to the so far described designs specifically intended for yeast [21, 22, 26] and bacterial cultures [23, 27]. Our system uses common algal model species Chlamydomonas reinhardtii, with the specific goal to use it as a herbicide resistance evolution model. C. reinhardtii is an established model species for herbicide resistance evolution [8, 28–30] and molecular analysis of herbicide resistance mutations [31–33], but all studies to date have used batch cultures. We present the full protocol for assembling and maintaining a 16-chamber mesostat array by a single person as well as control data illustrating the ability of the system to track trends and variability in the abundance of organisms among replicates. We also present pilot data illustrating the ability to use the mesostats to evolve resistance in C. reinhardtii to growth inhibiting herbicide glyphosate. Furthermore, we have included data from this

system illustrating the signal of common problems like leaks, contamination and cell clumping, showing how to distinguish it from biological variation as well as how to prevent and address these problems if they occur. We also outline the ways in which this system could be further modified and avenues of future research.

Discussion Content:

Lines 179–208: Chemostats offer a number of advantages over batch cultures for long-term experimental evolution research. Precise control of selective pressures in a chemically constant environment without evolutionary bottlenecks along with a link between growth rate and dilution rate constitute a useful conceptual framework for modelling evolutionary adaptation and population dynamics. This system adds to the small, but growing, number of efforts to produce simple but scalable, multiplexed DIY chemostats from cheaper materials that are possible to build and maintain by a single person [21–27], and is the first of its kind for experimental evolution of algae, specifically the evolution of herbicide resistance in model species C. reinhardtii. There are three substantive changes from the Miller et al. ministats [22], one system specific and two generic changes to suit experimental evolution with continuous sample extraction. Firstly the system was adapted to suit the study species C. reinhardtii, including light and a lower dilution rate, which distinguishes the system from previous DIY chemostat arrays developed for

maintenance of yeast [21, 22, 26] and bacterial cultures [23, 27]. Secondly, a needle and syringe system was added to facilitate easy, sterile access to the culture for the removal of samples. This allows sampling from the middle of the active culture rather than relying on the overflow. The efflux only samples from the top and the overflow chamber constitutes a wholly different environment without continuous dilution, build-up of waste products and increased evaporation, making them unrepresentative samples of the chamber populations. Furthermore this simplifies addition of cells or treatment compounds directly to the chambers, eliminating the risk of contamination that comes with disconnecting the medium influx or efflux channels. While sampling ports have been described before (e.g. [23, 25]) our simplification and combination with syringe extraction allows manual sampling with minimal contamination. The third change is an increase in the chamber volume to allow larger population sizes and possible future introduction of several trophic levels. Furthermore, this increases increases the amount of sample that can be extracted on a regular basis, extending the possibilities for the types of assays that can be performed to characterise evolution in action, as most of the

previous DIY chemostat arrays have been limited by their small working sizes [22, 23, 25]. Lastly, there were several changes to specific materials to lower the overall costs.

Furthermore, we have included additional pilot data demonstrating how the system may be applied to herbicide resistance evolution specifically, using growth inhibiting herbicide glyphosate. The added results have been pasted below (from lines 130–145):

Applicability to herbicide resistance evolution

Fig 2 shows the population densities of the four glyphosate treated populations and two control populations for 24 days following glyphosate treatment introduction. The glyphosate treated chambers exhibit population decline at a rate approximate to (150 mg/L, slope = -0.14, SE = 0.006) or below (100 mg/L, slope = -0.098, SE = 0.006) the dilution rate of 0.15/day. In the same timespan, the control populations exhibit an overall slight increase in population density (slope = 0.022, SE = 0.006), possibly reflecting adaptation to the mesostat environment. The onset of the population decline appears to be immediate for the 150 mg/L glyphosate treatment, whereas it occurs roughly 5 days after the glyphosate injection for the 100 mg/L glyphosate treatment. This is likely due to the 100 mg/L glyphosate treatment being just on the cusp of the minimum inhibitory concentration, enabling the populations to maintain growth for a short while before the herbicidal action is apparent. After 15 and 18 days respectively of population density decline, the 100 mg/L populations increase in cell density again, suggesting the populations have evolved resistance to the glyphosate, whereas the 150 mg/L populations never show evidence of resistance. 

The authors can be advised to set forth a scientifically relevant problem and solve it with the proposed mesostat(s) to demonstrate its applicability.

RESPONSE: Thank you for this suggestion. We have added additional data that provides insight into the nature of the relevant problems and questions that can be addressed via this protocol. 

We note that this work is a protocol via which three additional papers will be submitted.

Minor comments:

Introduction: Here I describe a… — please rephrase.

The authors mention “…this thesis..” in the section called “Limitations”, but this is an article, not a thesis, right?

RESPONSE: Thank you for noticing these mistakes, they have been corrected.

Reviewer #2: The manuscript describes a development of a low cost mesoscale continuous culturing system, which could be applied to investigate adaptive evolution of microalgae. The method has potential utility for research and describes some new developments, and the methods and materials are described in sufficient details for reproducibility. However some improvements are necessary for method validation and other points as indicated below.

line 77-78, replicability: it is not clear why only these dilution rates (0.15 and 0.3 /day) were applied. Is there a range of dilution rates, based on which the ideal dilution rate (i.e. to achieve steady state) could be established? Did the authors test some 'extreme' range of dilution rates to test the plasticity of the setup?

RESPONSE: Thank you for this comment. The chosen dilution rates were based on previous studies using similar organisms that this system was designed for, along with testing a lower dilution rate did not lead to major changes in the steady state population density, with the intention to decrease consumption of growth medium as well as wear and tear on the pump tubing. It was not intended to be a stress test. We have now addressed this in the Methods section (see pasted above in response to comment 3 by Reviewer #1 and on line 68–73 of the revised document).

Figure 2: the results of this experiment are not entirely clear, because there seems to be quite a large variations in treatments A-E as well, which did not undergo leak test. Conditions A-E need to be better explained.

Figure 3: There is some variability of population density in the control, non-contaminated cultures as well, so these results cannot be judged with certainty. Authors refer to 'likely' source of contamination, could not this be better verified with deliberately contaminating some of the cultures, e.g. by inoculating some bacteria?

RESPONSE: Thank you for these two comments. There are many examples that the variability we see in the unaffected populations here is normal for this type of system (e.g. Fussmann et al. 2000 Science. Crossing the Hopf Bifurcation in a Live Predator-Prey System; Yoshida et al. 2003 Nature. Rapid Evolution Drives Ecological Dynamics in a Predator-Prey System; Becks et al. 2012 Ecology Letters. The Functional Genomics of an Eco-Evolutionary Feedback Loop: Linking Gene Expression, Trait Evolution, and Community Dynamics. Note all are cited in the manuscript). 

While we cannot do more experiments to cause deliberate leaks or contamination, we believe the data presented in this manuscript from spontaneous faults to the system serves to effectively illustrate that the data signal of these errors, compared to normal variability within and among the unaffected populations, is very readily identified. 

To make this clearer we have highlighted the distinction between normal variability among replicates by highlighting variation in the replicability section as well as improving the description of the features of variation caused by faults in the common problems section (see changes in paragraphs pasted below). 

We have also added in clumping as a common problem based on your suggestion below, and pilot data for the population density response to the growth inhibiting herbicide glyphosate to demonstrate that the effect of a herbicide treatment is readily distinguishable from normal variation and equipment faults.

Changes copied from the revised document:

Among population variation, lines 123–129: The level of variation observed in this data set is normal for this type of system [16–18] and can be divided into among population variation and day-to-day variation. Among population variation is primarily caused by the biology of the system as these are separate, genetically heterogeneous populations on separate evolutionary trajectories. Day-to-day variation is however at least partly caused by limitations in sample processing. Both are discussed in more detail in the Discussion section, as well as how to reduce or circumvent the latter in particular.

Leak, lines 149–159: Compared to the expected among population and within-population day-to-day variation observed in the chambers that did not experience a leak, three crucial differences together make this the characteristic signal of over-dilution: 1) While similarly large day-to-day fluctuations in the measured density occur in the presented data set, day effects present across chambers. The rapid reduction in population density for chamber F between days 34 and 35 is only apparent in that chamber, whereas a similar reduction between days 37 and 38 is seen in all of chambers A–E. 2) The reduction in population density in chamber F results in a lower population density than otherwise observed in the data set (by roughly 3 x 105 cells/ml). 3) The reduced population density is observed in chamber F for several days after day 35, rather than recovering by the next day like seen for chambers A–E after day 38.

Contamination, lines 166–170: Furthermore, while there is considerable variation among all populations, the signal of contamination in the data is clearly distinguished from the expected among population variation and day-to-day variation by the fact that it is a consistent, long-term population-density decline without recovery 12 days after the contamination event.

Clumping, lines 174–177: The data signal here is an artefact of the limitations of the instrument being unable to accurately distinguish individual cells within aggregates, resulting in huge fluctuations in estimated cell density considerably larger than and out of step with the day-to-day variation observed in the other populations.

Finally, as sources of variation are of pivotal importance in ecology and evolution studies, a subsection has also been added to the discussion section on sources of variation, highlighting those where it cannot be minimised as well as suggesting methods to reduce (pasted below from lines 210–292):

Sources of variation and how to minimise it 

The data presented here illustrates the expected variation between cultures and how to identify the signal of equipment failure, such as a leak, or contamination. We also demonstrate that the system can be used to evolve resistance to growth inhibiting herbicide glyphosate, and that the signal of herbicidal action is apparent as a population density decline, followed by an increase after the population has evolved resistance. The herbicidal effect is clearly distinguishable from the expected variation under control conditions, and given enough time, the resistant population is expected to settle at a new steady state.

The variation among replicate populations observed here is normal [16–18] and expected as they constitute separate, genetically heterogeneous populations on separate evolutionary trajectories. Even when using a single founder population, the genetic bottleneck caused by splitting it between populations as well as the dynamics during the establishment phase of batch-like growth dynamics [6] will result in similar but distinct populations by the time they reach steady state. Effort should be made to ensure that all chambers receive the same levels of light and aeration as well as consistent dilution with the same medium, and starting variation could be eliminated through starting with clone populations at a high enough concentration to effectively avoid the establishment phase. However, the among population variation is generally of scientific interest to experimental evolution studies and should be investigated rather than eliminated.

Conversely, while day-to-day variation within a population is also normal for this type of system, it is also partly caused by limitations to the sampling protocol. The data presented here was obtained from measurements performed on living cells that had the opportunity to grow and divide between sample extraction from the mesostat chambers and sampling processing. While this is an unavoidable source of variation, it can however be reduced by minimising the time that passes and working in a controlled temperature environment. If the experimental design allows, the cells can be immobilised by using e.g. Lugol’s solution before counting with flow- or haemocytometry. It is also possible to control for this variation by including sampling day as a source of error in statistical models applied to the data

The among population and day-to-day within population variation are however both clearly distinguishable from the data signal of common faults like leaks, contamination and clumping. While these faults are likely to be detected before they become apparent in the population density data, leaks causing significant over-dilution are apparent within a few hours while clumping and contamination can be observed under a microscope, it is important to understand how they affect the data so that an informed decision can be made on how to handle it. While the population density is always expected to quickly return to steady state after over-dilution, the increased flushing out of cells constitutes an evolutionary bottleneck and the changed growth conditions may affect other traits of the population not visible in the population density data and data collected subsequent to a major leak should thus be treated with caution. The leak presented here was caused by equipment failure resulting in over-dilution, but smaller leaks often occur as the pump tubing wears out with long term use, which can lead to under-dilution of the connected chambers. Both are best prevented by regular inspection of the pump parts for irregularities

Bacterial contamination is another common risk in long-running continuous cultures [15], and is best prevented by working in a sterile environment and minimising the points at which contamination can enter the system. The main contamination risk presents when disconnecting any part of the array, such as when switching medium containers, or when extracting samples, and particular care should be taken to keep the connecting parts sterilised during. The example presented here is the only instance of contamination observed across eight separate experiments each lasting more than a month and happened when the chambers were disconnected from the array for a longer period of time and removed from the sterile environment. Even so, only four out of six chambers showed evidence of contamination under the microscope 12 days after the contamination event, despite all of the chambers in question sharing a medium source. This suggests that the system is robust in terms of contamination not spreading between the chambers. While regular microscopy inspection of cell samples for contamination is recommended, this can be laborious with a large number of replicates and the characteristic population density decline provides another opportunity to detect and isolate the problem.

Lastly, chemostat populations being under a selective pressure to evolve phenotypes reducing their risk of being flushed out is an often cited issue with the method [7, 15], presenting as adhesion to the chamber walls and cell flocculations. While this phenomenon has as of yet never been observed under control conditions with this protocol, there was one instance of cells exhibiting a clumping phenotype while under treatment with a sublethal dose of glyphosate, making it possible it was a response to the treatment rather than the mesostat environment. In C. reinhardtii there are two distinct types of clumping: cell aggregations of separate cells and palmelloid colonies that share a cell wall [40–42]. Both have been found to be an induced response as well as heritable [40, 41, 43–45], meaning that once they become common in a population they may be hard to get rid of [40]. Palmelloids are small enough that they will not cause blockages, but due to the shared cell wall they cannot be disassociated through bubbling or by vortexing a sample. Cell aggregations can be considerably larger, however they are also possible to break apart through vortexing, and vigorous bubbling of the cultures often prevents their formation [7]. How much of a problem clumping is depends on the experiment, i.e. it becomes a problem if it hinders sample processing and when it is thought to be an artefact of the chemostat environment rather than in response to the applied treatment. For population density measurements by flow cytometry as presented here, clumping considerably reduces the accuracy of the measurements as each clump is counted as a single particle, increasing day-to-day variation. In this case, manual haemocytometry could give a better estimate but this is considerably more laborious.

Besides the common problems investigated, are there any potential problems with the introduced mesoscale chemostat system? For example changes in system pressure, aeration, problems with cell clumping, clogging of the tubing system and needles, etc. The impact of these potential problems should be mentioned and possible solutions offered to circumvent these issues.

RESPONSE: Thank you for this idea. The prevention of problems relating to aeration and blockages is addressed by the Daily Maintenance section of the protocol, and we have expanded the detail around this (pasted below). We have also included this set of issues and solutions in the Discussion (see pasted below). Furthermore, we have extended the Common Problems section to include representative data for the data signal of clumping. This is also addressed in the Discussion and linked to detecting variation due to protocol issues vs. biological variation (see above).

Protocol, Step 18.2 Check culture chamber airflow is satisfactory and equal. If a culture chamber has low airflow, check if the filters are blocked, first the filter on the adjoining collection chamber, then the filter connecting to the manifold, and replace if necessary. The most common cause of filter blockage is them becoming wet, either by an efflux blockage or low pressure resulting in the culture entering the aeration needling and tubing, or by high ambient humidity. Make a note of airflow problems data is being collected if it is possible the problems have been present for more than an hour. If filters are often blocked, consider changing the ambient humidity.

Protocol, Step 18.4 Check culture levels are even and do not deviate from the 380 ml line. If the medium level is too low, check the media influx tubing and needle for blockages. The most likely points for blockages are inside the pump tubing and any needles due to their narrow gauges. If medium influx is normal, adjust the efflux needle, and check if the level is back to normal in a couple of hours (time needed to wait dependent on flow rate and total volume deviation). If the level is too high, examine the efflux tubing and needle for blockages, along with the collection chamber filter. When unblocked, the culture chamber level should return to normal volume relatively quickly. Always make a note of culture level changes if data is being collected.

Discussion, lines 316–327: Another potential problem involves insufficient aeration or efflux blockages causing over- or under-pressure in the chambers. Provided the air supply is sufficient, the most common reason for low or uneven bubbling is blocked air filters, usually because they have become damp. If the air filters frequently become damp, it may be due to too high ambient humidity. Not enough bubbling may cause sedimentation and stratification of the culture, as well as selection for phenotypes that sink so they avoid being flushed out, or it may instead cause the culture to rise through the aeration needle instead of through the efflux needle. Clogging of the media line is uncommon, but can occur if not properly sterilised and contamination is allowed to grow. This is often apparent as a reduction in flow into and out of the affected chambers. The daily maintenance part of the protocol outlines how to spot and address these problems. 

line 39: perhaps plural pronoun should be used throughout, as there are multiple authors of the work.

RESPONSE: We have made this change. Thank you for pointing this out.

line 118-119 - if the work was the basis of a thesis, add the relevant reference of the thesis.

RESPONSE: See response to Reviewer #1 above — this work is the first paper submitted for publication from a thesis, the three papers that have utilised this system are currently in preparation for publication before the end of the year.

line 128-129: 'the system must therefore be carefully balanced against the resulting biology' -this is unclear, please clarify/define this more precisely.

RESPONSE: Thank you for this comment. This statement was referring specifically to cost saving measures like lowering the dilution rate also changing the selective pressure experienced by the population. We have edited the relevant section to make this clearer (pasted below from lines 302–307):

One way to conserve medium and treatment components is to lower the dilution rate, which in the experiments presented here had no effect on population density in the chambers. However, this changes the selection pressures experienced by the populations as well as their doubling rate [15]. The logistics of the system and any cost saving measures must therefore be carefully balanced against the resulting biology, taking into account the desired selective pressure, cell cycle stage and generation time.

line 133-135: are there pilot experiment performed on the potential effects of sampling frequency on population density and potential disruption of the system?

RESPONSE: Thank you for this idea. Unfortunately we have no data available on this, it is an assumption made from theory – the constancy of the environment and thus the selective pressure is based upon continuous dilution. However, see the referenced Fischer et al. 2014 (“The Exponentially Fed Batch Culture as a Reliable Alternative to Conventional Chemostats”) as this is something they try to address with their design.

line 148-150: In a high performance chemostat system, these sensors are indeed important, therefore the authors need to demonstrate the feasibility of introducing these sensors in their system.

RESPONSE: There is a notable example, the “omnistat”, fitting many sensors into a DIY system (Ekkers et al. 2020. The Omnistat: A Flexible Continuous-Culture System for Prolonged Experimental Evolution). Our mesostats are however an example of a simpler DIY system where sensors are an optional extra, afforded by the fact that it is overall larger and the lids thus have the space for them if needed for the experiment in question. The text has been amended to reference this paper and clarify this detail (pasted below from lines 340–341):

As the chamber lids are relatively large, sensors to monitor e.g. pH or CO2 levels could also be fitted through additional ports (see [25]). 

The light system lacks some details. What was the light source applied? Could the authors provide the emission spectrum of the lamp? Is it possible to provide diurnal light cycles? Is it possible to change the spectral composition of the light source? So could it be tuned for different photosynthetic organisms with different photosynthetic pigment or light harvesting antenna composition?

RESPONSE: Thank you for these ideas. The light sources are addressed in the overview of the design in the protocol: “When growing photosynthetic organisms such as algae, even light levels for all chambers are best maintained by a light table as well as fitting strip lights around the chambers.” This has been extended in the Overview of the design section of the protocol (see pasted below) and the lights have been added to the Materials list. The current light system design is very simple, as that is sufficient for C. reinhardtii and the questions we used the system to address, but the majority of your suggested modifications would indeed be possible to implement, and we have now enhanced this detail in the How to modify or improve section (see pasted below).

Overview of the design: The light system

The light is provided by LED strip lights mounted around the chambers and between the two rows of chambers, as well as a DIY light box consisting of LED strip lights and a semi-transparent plastic top to diffuse the light. Equal light from all angles is essential to ensure even algal growth in the chambers. A light box is not necessary, but convenient and can be used for providing light to batch cultures or growth assays of subsamples.

Discussion, lines 352–361: The light system here is rudimentary but sufficient for C. reinhardtii growth [40], using warm white light LED strip lights mounted around the chambers along with DIY light box also consisting of white light LED strip lights and a semi-transparent plastic top to diffuse the light. The light box is not necessary, but convenient for maximising light from all angles. Under control conditions 24h light was used, but it is possible to fit a timer to the outlet connecting the lights to instead provide a diurnal light cycle. Coloured semi-transparent plastic could be used to provide light only from a specific part of the light spectrum, but it would also be possible to mount specialist lighting around the mesostats if tuning for a specific photosynthetic organism or experiment is desired.

In the Discussion, authors describe several potential future improvements and research opportunities, but how feasible are these practically for the introduced system? For example, temperature control seems to be unsolved for the current stage of development, and the application of e.g. a water bath may need a significant redesign of the entire system. The feasibility of the proposed applications and future developments should be compared and analyzed with other existing continuous culturing systems.

RESPONSE: Thank you for this comment. We appreciate this request which we interpret as a request for the context and potential of these types of platforms for a variety of experimental evolution work. We have increased our discussion of the context of our work by comparing it to other protocols (see new Discussion lines 179–208, pasted above in response to reviewer #1), and identified a few ways to extend our protocol with minimal modification (see pasted below).

Lines 346–351: Any water-tight, sterilisable container can be used for culture chambers if suitable lids can be manufactured, such as falcon tubes [23] or commonly available lab glassware [25]. The controlled temperature room can be replaced with water baths (note however that this requires mounting the lights up on the sides of the water baths), and portable aquarium pumps can be used instead of building infrastructure, increasing flexibility in where the system can be housed.

Lines 356–361: Under control conditions 24h light was used, but it is possible to fit a timer to the outlet connecting the lights to instead provide a diurnal light cycle. Coloured semi-transparent plastic could be used to provide light only from a specific part of the light spectrum, but it would also be possible to mount specialist lighting around the mesostats if tuning for a specific photosynthetic organism or experiment is desired.

Lines 363–375: We have used this system for experimental evolution of herbicide resistance in algae by adding glyphosate as a shock injection and then continuously through the growth medium, however, this setup is also easily adaptable depending on the research question. The herbicide treatment could also be applied gradually through the medium or through series of shock injections in a ratchet protocol [28] and investigate to what level the resistance can be pushed and at what speed. The dilution rate and thus the cell growth rate is set by the pump speed, tubing thickness and culture volume, so running chambers with different dilution rates simultaneously would be possible with different pump tubing thicknesses, multiple pumps or multiple culture volumes, depending on the range required. Furthermore, the use of multiple light tables with opaque partitions between cultures would allow testing for an interaction with light level, or the chambers could be kept in water baths at different temperatures to determine the effect of temperature.

For example, the multiple trophic level experiments referenced were all single chemostat systems, and our system was specifically designed for the same types of populations with regards to size and light levels but allowing simultaneous replication. This is now made clear in the closing paragraph (excerpt from lines 381–387 pasted below).

Lines 342–345: The predator-prey cycles of rotifer Brachionus calyciflorus and C. reinhardtii as well as Chlorella vulgaris have been successfully studied and modelled using chemostat environments (e.g. [17,18]) and our setup allows simplified simultaneous replication for this type of system that can be maintained by one person. Competition could also be introduced to the system through using multiple algal strains and monitoring their frequencies or through expanding the culture ecosystem to include other algal species or bacteria [48].

With regards to temperature control, as the system is waterproof, introducing water baths for temperature control is as simple as placing the chambers in a water bath. Strip lights can be mounted out of the water on the sides and above, which is something we have done in the past for batch cultures. We have now clarified in the text that minor adjustments to light placement would be required (see above or on line 350).

7. PLOS authors have the option to publish the peer review history of their article (what does this mean?). If published, this will include your full peer review and any attached files.

Do you want your identity to be public for this peer review? For information about this choice, including consent withdrawal, please see our Privacy Policy.

Reviewer #1: No

Reviewer #2: No

---

## [Decision Letter · Decision Letter 1]

13 Jul 2022

Mesostats — A multiplexed, low-cost, do-it-yourself continuous culturing system for experimental evolution of mesocosms

PONE-D-22-12361R1

Dear Dr. Hansson,

We’re pleased to inform you that your manuscript has been judged scientifically suitable for publication and will be formally accepted for publication once it meets all outstanding technical requirements.

Kind regards,

Maya Dimova Lambreva, Ph.D.

Academic Editor

PLOS ONE

Additional Editor Comments (optional):

Reviewers' comments:

Reviewer's Responses to Questions

**Comments to the Author**

1. Does the manuscript report a protocol which is of utility to the research community and adds value to the published literature?

Reviewer #1: Yes

Reviewer #2: Yes

2. Has the protocol been described in sufficient detail?

Descriptions of methods and reagents contained in the step-by-step protocol should be reported in sufficient detail for another researcher to reproduce all experiments and analyses. The protocol should describe the appropriate controls, sample sizes and replication needed to ensure that the data are robust and reproducible.

Reviewer #1: Yes

Reviewer #2: Yes

3. Does the protocol describe a validated method?

Reviewer #1: Yes

Reviewer #2: Yes

4. If the manuscript contains new data, have the authors made this data fully available?

Reviewer #1: Yes

Reviewer #2: Yes

**5. Is the article presented in an intelligible fashion and written in standard English?**

Reviewer #1: Yes

Reviewer #2: Yes

6. Review Comments to the Author

Reviewer #1: In my opinion, the authors had dealt with the issues rised in a satisfactory manner, so they contribution can be published now.

Reviewer #2: The authors addressed the comments satisfactorily and the revised manuscript has significantly improved in my opinion.

7. PLOS authors have the option to publish the peer review history of their article (what does this mean?). If published, this will include your full peer review and any attached files.

Reviewer #1: **Yes: **Alexei Solovchenko

Reviewer #2: No

---

## [Editor Report · Acceptance letter]

18 Jul 2022

PONE-D-22-12361R1 

Mesostats — A multiplexed, low-cost, do-it-yourself continuous culturing system for experimental evolution of mesocosms 

Dear Dr. Hansson:

I'm pleased to inform you that your manuscript has been deemed suitable for publication in PLOS ONE. Congratulations! Your manuscript is now with our production department. 

Kind regards, 

on behalf of

Dr Maya Dimova Lambreva 

Academic Editor

PLOS ONE